# Investigation of Clofazimine Resistance and Genetic Mutations in Drug-Resistant *Mycobacterium tuberculosis* Isolates

**DOI:** 10.3390/jcm11071927

**Published:** 2022-03-30

**Authors:** Sanghee Park, Jihee Jung, Jiyeon Kim, Sang Bong Han, Sungweon Ryoo

**Affiliations:** 1Clinical Research Center, Masan National Tuberculosis Hospital, Changwon-si 51755, Korea; loody23@gmail.com (J.J.); jyeon9127@korea.kr (J.K.); viweon@gmail.com (S.R.); 2Department of Laboratory Medicine, Masan National Tuberculosis Hospital, Changwon-si 51755, Korea; astrobow@gmail.com

**Keywords:** clofazimine, mutation, resistance, MDR, XDR, mycobacterium, tuberculosis

## Abstract

Recently, as clofazimine (CFZ) showed a good therapeutic effect in treating multi-drug-resistant tuberculosis (MDR-TB), the anti-tuberculosis activity and resistance were re-focused. Here, we investigated the CFZ resistance and genetic mutations of drug-resistant *Mycobacterium tuberculosis* (DR-Mtb) isolates to improve the diagnosis and treatment of drug-resistant TB patients. The minimal inhibitory concentration (MIC) of CFZ was examined by resazurin microtiter assay (REMA) with two reference strains and 122 clinical isolates from Korea. The cause of CFZ resistance was investigated in relation to the therapeutic history of patients. Mutations of Rv0678, Rv1979c and pepQ of CFZ resistant isolates were analyzed by PCR and DNA sequencing. The rate of CFZ resistance with MIC > 1 mg/L was 4.1% in drug-resistant Mtb isolates. The cause of CFZ resistance was not related to treatment with CFZ or bedaquiline. A CFZ susceptibility test should be conducted regardless of dugs use history. The four novel mutation sites were identified in the Rv0678 and pepQ genes related to CFZ resistance in this study.

## 1. Introduction

The CFZ, a fat-soluble riminophenazine dye-based antibiotic, has both antimycobacterial and anti-inflammatory activities [1,2]. The action of CFZ has suggested that the outer membrane is its primary action site, and the respiratory chain and ion transporters are the putative targets [3,4]. CFZ has shown activity against Mtb, including multidrug-resistant strains in vitro and in animal studies [5,6]. However, the inconsistency of therapeutic effects in animal infection experiments did not make them a choice for TB treatment [7,8]. In addition, CFZ has a relatively low drug concentration in plasma, skin, and gastrointestinal side effects and a long drug half-life [9,10]. It has not received attention as a treatment for TB as an oral drug. The interest in a CFZ-containing therapeutic regimen for TB has been re-focused after a study showed that a regimen containing CFZ and other drugs including high-dose fluoroquinolones was very effective against MDR in Bangladesh [11]. A systematic review presented observational studies and found that after treatment with CFZ, those with MDR and XDR TB experienced favorable treatment outcomes [12]. CFZ has recently attracted interest in DR-TB management, which is one of the major public health problems with TB treatment. According to the TB treatment guidelines that the WHO revised in 2016, CFZ, which was a group 5 drug, was newly included in the core drugs as it was classified as group C of the second drug [13]. A meta-analysis concluded that MDR-TB patients’ treatment outcome was significantly better when using linezolid, a later generation of fluoroquinolone, bedaquiline, CFZ and carbapenems, compared with standard treatment [14]. CFZ was added as a second-line drug and administered to MDR patients for more than nine months in multi-country clinical studies [12,13,14]. However, CFZ’s adverse effects of skin pigmentation occurred in most patients [12]. The study of the distribution of CFZ resistance to Mtb is needed for patients’ benefits and risks.

A breakpoint for CFZ susceptibility testing has not been defined by the Clinical & Laboratory Standards Institute (Wayne, PA, USA) or US Food and Drug Administration (Silver Spring, MD, USA), although the WHO has tentatively determined a critical concentration of 1 mg/L in MGIT based on small studies and unpublished data [15]. Previous studies have demonstrated that CFZ has MICs for Mtb, including multidrug-resistant strains, typically ranging from 0.125 to 2.0 mg/L [16,17,18]. Some studies have recommended CFZ breakpoints that range from 0.25 to 1 mg/L [19,20]. The resistance of CFZ supporting laboratory and clinical evidence is considered insufficient [21,22].

Resistance-associated variants (RAV) that lead to increased MICs of CFZ have been described in three genes of Mtb: Rv0678, Rv1979c and pepQ [23,24,25]. RAVs in Rv0678, a gene regulating the expression of the MmpS5-MmpL5 efflux pump, lead to 2- to 4-fold increases in CFZ MIC [23,26]. They have been isolated in vitro upon exposure to CFZ. However, the small number of clinical isolates related to CFZ resistance with MICs > 1 mg/L were reported in previous studies [24,27,28,29]. The mutations in Rv1979c, encoding a probable amino acid membrane transporter with permease activity, were reportedly associated with CFZ resistance in vitro [23] and in a clinical isolate with CFZ MICs > 1.2 mg/L [24]. The mutations in the putative proline aminopeptidase gene pepQ have been shown to confer low-level cross-resistance between bedaquiline and CFZ in vitro [25].

This study investigates 122 drug-resistant clinical isolates and two reference strains of Mtb of CFZ resistance. We determined the range of MIC to CFZ and investigated the genetic variability with CFZ resistance to improve for diagnosis and treatment of drug-resistant TB patients.

## 2. Materials and Methods

### 2.1. Clinical Isolates of Mtb

The 82 MDR and 40 XDR isolates were supplied from the Tuberculosis Specimen Bank of Masan National Hospital. The clinical isolates were collected from November 2009 to January 2016. TB that is resistant to both isoniazid and rifampin, which are representative primary anti-TB drugs, is defined as MDR. Among them is one resistant to one of the injectable drugs and fluoroquinolone drugs, which is defined as XDR [26]. The definition of XDR before 2021 was used for drug resistance criteria. The isolates used in this study did not have drug susceptibility test results for bedaquiline or linezolid. The clinical isolates of Mtb were sequentially cultured in Middlebrook 7H9 Broth (BBL^TM^ MGIT ^TM^ Mycobacteria Growth Indicator Tube, Becton-Dickinson, Sparks, MD, USA) and Ogawa II agar (Asanpharm, Seoul, Korea). The bacterial colonies cultured 21 to 26 days on the agar were taken and applied to determine MIC, PCR, and DNA sequencing.

### 2.2. Determination of MIC

The CFZ MIC of Mtb was measured using the resazurin microtiter assay (REMA) because it is inexpensive, rapid, and simple to perform [30]. Resazurin is an oxidation–reduction indicator. A change from blue to pink indicates a reduction of resazurin and therefore bacterial growth. The REMA plate method was performed in 7H9-S broth containing Middlebrook broth, 0.1% Casitone, and 0.5% glycerol and supplemented with oleic acid, albumin, dextrose, and catalase (Becton-Dickinson, Sparks, MD, USA). CFZ (Sigma-Aldrich, St. Louis, MO, USA) was added to 7H9-S broth, with final concentrations ranging from 0.031 to 16 mg/L in 96-well plates for convenience of results reading. Growth controls containing no antibiotic and sterility controls without inoculation were also included. The Mtb H37Rv (ATCC 25618) and Mtb K (NCCP 15986) were used as reference strain validation of REMA. The inoculum was adjusted to McFarland 1.0 with DensiCHEK^TM^ plus instrument (bioMereux, Marcy-l'Étoile, France) from fresh colonies on Ogawa II agar and further diluted 1:10 in 7H9-S broth, and 100 μL was used as an inoculum. The plates were covered, sealed in plastic bags, and incubated at 37 °C in the normal atmosphere. After 7 days of incubation, 30 μL of resazurin (Fisher scientific, Waltham, MA, USA) solution was added to each well, incubated for up to 48 h at 37 °C, and assessed for color development. The MIC was defined as the lowest drug concentration that prevented this color change. The resistance criterion for CFZ was read as >1 mg/L [20,23]. Drug susceptibility tests for the isolates showed MIC > 0.5 mg/L in CFZ and were repeated twice.

### 2.3. Statistical Analysis

The comparison of resistance of MDR and XDR isolates to CFZ were performed using the one-sample *t*-test. A cross-tabulation analysis was used to assess CFZ resistance between MDR and XDR isolates. The MICs were recorded as discrete ordinal values and were not continuous, and resistant numbers of isolates were small, so we used the nonparametric test. The paired-sample *t*-test was used to compare the MIC_50_ and MIC_90_ of MDR and XDR isolates to CFZ. A *p*-value of <0.05 was considered statistically significant. All statistical analyses were performed using SPSS Statistics version 20.0 (IBM, Armonk, NY, USA).

### 2.4. Clinical Data Analysis of CFZ-Resistant Isolates

The clinical information was invested by medical staff in electronical medical records to analyze the relationship of treatment history and progress to patients with the CFZ resistance acquisition of the Mtb isolates.

### 2.5. PCR and DNA Sequencing

Mtb DNA was extracted using a commercial kit, DNeasy^TM^ UltraClean^TM^ Microbial Kit (Qiagen, Hilden, Germany) for PCR of the CFZ resistance-related genes. PCR amplification of the genes were performed using Maxime™ PCR premix i-StarTaq (iNtRON, Seongnam, Korea). The two reference strains and five isolates with a CFZ MIC of >1 were selected, and these DNA were extracted to invest the CFZ resistance gene mutation. The Rv0678, Rv1979c, and pepQ genes were amplified by PCR, respectively. The genomic DNA from CFZ-resistant mutants was isolated and reference strains were subjected to PCR amplification using Rv0678 primers rv0678F (5′-TGCCTTCGGAACCAAAGAA-3′) and rv0678R (5′-GACAACACGGTCACCTACAA-3′) as described previously [23] The Rv1979 gene was PCR-amplified using primers rv1979cF (5′-GCGGCGGAAATGAGTGT-3′) and re1979cR (5′-ATGCACGACGGCTTTATCA-3′) [23]. The gene was PCR-amplified using primers pepQF (5′-ATCAATGCCCCCTGGAAC-3′) and pepQR (5′-GCAGTTCTTCAACTTGGTG-3′) [25]. The PCR products were sequenced with the same primers used for amplification by Bioneer (Daejeon, Korea). For analysis of the sequences, ClustalW analysis of Mega 10.2.5 software was used to align and compare resistance-related gene sequences [31,32]. The gene polymorphisms were identified by aligning with the reference strain H37Rv (GenBank ID: NC_000962.3).

## 3. Results

### 3.1. CFZ MIC to Mtb

The CFZ’s MIC value of the H37Rv used as a reference strain was 0.25 to 0.5, and K was from 0.13 to 0.25, respectively. The distribution of Mtb isolates at the MIC of CFZ shown in Figure 1. Among the 122 isolates consisting of 82 MDR and 40 XDR, the number of bacteria showing MIC > 1 mg/L as determined by CFZ resistance was 4.1%. The CFZ resistance rate was 1.2% in MDR and 10.0% in XDR isolates, respectively. The CFZ resistance rate was higher in XDR than in MDR isolates (*p* = 0.001). CFZ’s MIC_50_ and MIC_90_ values, which mean 50% and 90% inhibition of the target bacteria, in all isolates were 0.13 and 0.25, respectively. The MDR CFZ’s MIC_50_ and MIC_90_ were 0.13 and 0.25, respectively. The XDR CFZ’s MIC_50_ and MIC_90_ were 0.25 and 0.5. The CFZ’s MIC_50_ values and MIC_90_ values were not statically significant (*p* = 0.09 and *p* = 0.11, respectively).

### 3.2. Clinical Data Analysis of CFZ Resistant Isolates

Drug resistance data of clinical Mtb isolates are shown Table 1. The five CFZ-resistant Mtb were isolated from four patients between 2010 and 2014. A patient was diagnosed with MDR TB, and the other three patients were diagnosed with XDR TB. The treatment history of patients and bacterial culture test of isolates are shown Figure 2. The P1 patient with MDR isolates was treatment-completed. The P2 was a chronic excretor, patients of sputum AFB smear-positive and culture-positive after treatment and who died from treatment failure. The P3 patient was a chronic excretor and was in treatment failure. The P3 patient had a history of arbitrarily stopping taking CFZ for about eight months and bedaquiline (BDQ) for six months during inpatient treatment. However, the two isolates were cultured from the patient before administering CFZ and BDQ. The P4 patient was loss to follow-up. The three patients had no history of taking CFZ and BDQ during hospitalization.

### 3.3. Sequencing of Genes Related to Resistance

The mutations of CFZ resistant isolates were observed in Rv0678 and pepQ in this study. The results showed that no mutation was observed in Rv1979c among all the isolates. The CFZ resistance-related gene mutation sites and related isolates are shown in Figure 3. The G138 insertion in the Rv0678 gene, known as an efflux pump Mmpl5 suppressor, was observed in CFZ-resistant isolate 9199. The G149A mutation in the Rv0678 gene, was observed in two CFZ-resistant isolates, 21873 and 22049. The mutation of G415T of pepQ known as cytoplasmic petidase was observed in isolate 22018. The mutation of G1010T was observed in isolate 9199. The G138 insertion of Rv0678 and G1010T of pepQ were observed from the same isolate 9199. The mutation sites of isolate 22033 were not detected in CFZ resistance-related genes. The 21873 strain is MDR. The 22033 and 22049 strains were all resistant to rifampin with XDR, and the sensitivity to bedaquiline could not be confirmed in this study. The 22018 and 22033 strains were isolated from the same patient. This patient had a history of arbitrarily stopping taking CFZ for about 6 months during inpatient treatment. However, as a result of administering CFZ after the bacteria was isolated, the association between drug discontinuation and Mtb mutation is expected to be low.

## 4. Discussion

One of the major problems with TB treatment is the emergence of MDR-TB and XDR-TB. Recently, the WHO has updated the definition of XDR-TB [33]. The new definition of pre-XDR is TB that meets MDR and rifampicin-resistant TB requirements and is resistant to fluoroquinolone. XDR meets MDR-TB and is resistant to fluoroquinolone and is resistant to one or more additional group A drugs, bedaquiline or linezolid. This leads to adequate access to treatment options for patients with resistant TB. There were no drug susceptibility test results for bedaquiline and linezolid in this study, so it is difficult to apply the latest accurate XDR bacteria. However, using the fluoroquinolone drug test results of isolates, most of the XDR isolates in this study correspond to pre-XDR. Through this study, CFZ resistance is suspected to be caused by drugs other than bedaquiline or CFZ, so it is necessary to conduct a drug susceptibility test before deciding to treat pre-XDR patients with CFZ for preventing adverse events from unnecessary drug administration.

A breakpoint concentration of 1 mg/L for CFZ susceptibility testing using the MGIT 960 method was proposed after the study of 26 multidrug-resistant clinical isolates revealed an MIC_90_ of 0.25 mg/L [15]. Previous studies have demonstrated that CFZ has MICs for Mtb H37Rv that typically range from 0.5 to 1.0 mg/L [18,20]. The in vitro resistance study surveyed CFZ MIC values for 90 clinical isolates mostly from patients with XDR-TB. The five (5.6%) isolates with MICs of >1.2 mg/L were determined as CFZ-resistant [20]. The five (5.6%) CFZ-resistant strains determined >1.2 mg/L isolated from 80 drug-resistant isolates and 10 drug-susceptible isolates [24]. The other previous study demonstrated that 27.7% of the 195 MDR-TB isolates were CFZ-resistant with MICs of >1 mg/L [34]. The serum peak of CFZ is 0.369 mg/L after a single 200 mg dose administered with anti-TB medicines and a high-fat meal and 1.0 mg/L average serum concentration for 300 mg daily dosing [35,36]. The MIC for CFZ of H37Rv observed in this study was similar when compared with previous research reports. The rate of CFZ resistance with MIC > 1 mg/L was 4.1% in DR-Mtb isolates in this study. The CFZ resistance rate was higher in XDR than in MDR isolates. The CFZ rates of MDR and XDR observed in this study were different when compared with previous research reports. The results of this study could be referred to as defining the breakpoint concentration for CFZ susceptibility testing.

The rv0678 gene, known as transcription repressors of the efflux pumps MmpL5 and MmpS5, has been reported to be related to drug resistance to rifampin, bedaquiline, and CFZ [20,23,24,26,29]. The CFZ-resistant isolates in this study did not have the same positions of mutation of Rv0678 gene’s previous reports, the sites of T2C, G193 deletion, G193 insertion, C466T, A202G, and C364 insertion. Although the site of resistance mutation is different from previous reports, mutations at different positions in the Rv0678 gene mean different amino acid expression, which is expected to be involved in CFZ resistance by increasing efflux pump expression.

The G265T, T157C, two frameshifts at codons 14 and 271 mutation sites of the pepQ gene, known as cytoplasmic petidase, presumed that resistance was mediated by a loss of function mutations [23,24,25]. The Leu to Ile at codon 145 of the pepQ gene mutations related to bedaquiline and CFZ resistance have been reported in clinical isolates [37]. We could not observe the same position of pepQ mutations of previous studies in the CFZ-resistant isolates in this study. However, mutations of G415T and G1010T were observed in our isolate. It is expected that inactivation of the protein expressed by the pepQ gene and changes in the amino acid structure will occur. The function of pepQ is to inhibit drug efflux, so it is presumed that the occurrence of mutations in this region increases the efflux pump function. Further studies such as efflux pump function and related drugs are needed to elucidate the mechanism of CFZ resistance.

Rv1979c is a putative permease that might be involved in amino acid transport [23,24]. Rv1979c might be involved in CFZ transport or uptake, either directly or indirectly, to alter the physiology of the bacteria to be less susceptible to the effect of CFZ. No mutation was detected in Rv1979c among CFZ resistant isolates in this study.

Although an isolate 22033 strain was resistant to CFZ, no mutation sites were found in the Rv0678, Rv1979c, and pepQ genes. It is necessary to elucidate the cause of resistance of this isolate through additional research such as whole-genome sequencing analysis.

Our study has too few CFZ-resistant strains and the retrospective nature of the analysis limits our conclusion of the molecular determinants of resistance to CFZ phenotype. The clinical CFZ-resistant strains have not been commonly observed in previous clinical studies [20,34]. The other studies have found some mutations potentially associated with resistance at lower MICs with >1 mg/L [25,29]. Testing of a larger number of strains from multicenter work is needed to conclusively identify a breakpoint value for CFZ with the REMA method and fully understand the association of CFZ resistant gene mutations with distinguished CFZ MICs.

## Figures and Tables

**Figure 1 jcm-11-01927-f001:**
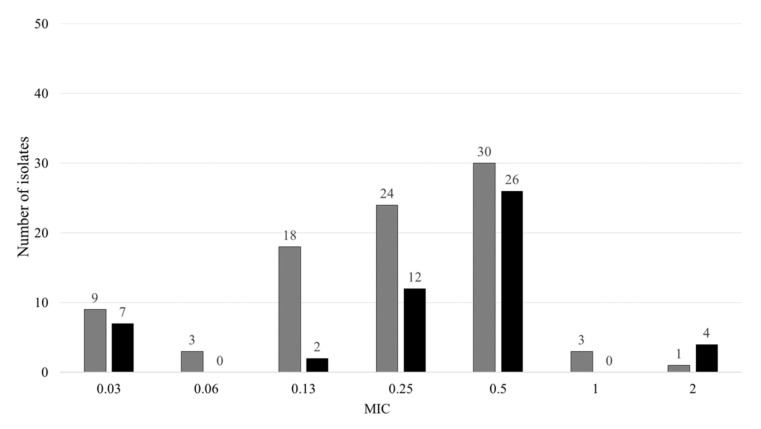
Distribution of minimum inhibitory concentration (MIC, mg/L) of CFZ for MDR (*n* = 82, gray bar) and XDR (*n* = 40, black bar). The resistance criterion for CFZ was read as >1 mg/L. The CFZ resistance rate was higher in XDR than in MDR isolates (*p* = 0.001).

**Figure 2 jcm-11-01927-f002:**
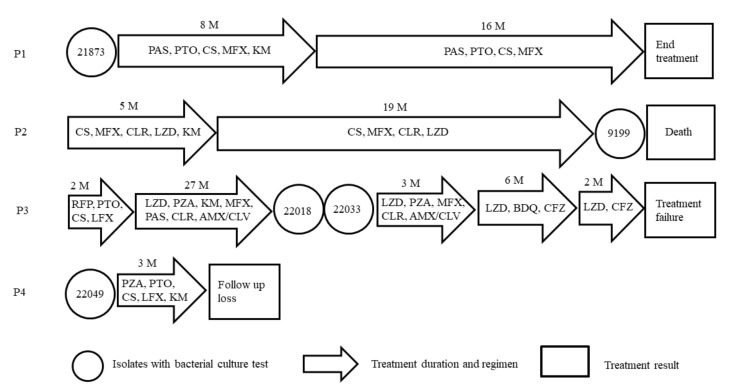
The treatment history and isolates of patients. M, month; AMX/CLV, amoxicillin/clavulanate; BDQ, bedaquiline; CFZ, clofazimine; CLR, clarithromycin; CS, cycloserine; KM, kanamycin; LZD, linezolid; MFX, moxifloxacin; PAS, para-aminosalicylic acid; PTO, prothionamide; RFP, rifampicin; PZA, pyrazinamide.

**Figure 3 jcm-11-01927-f003:**
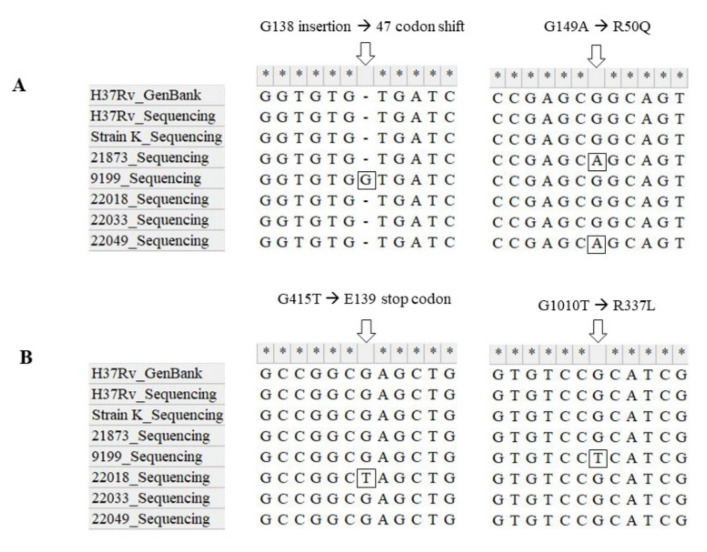
CFZ resistance-related gene mutation sites and related isolates and reference strains. (**A**) Rv0678 gene; (**B**) pepQ gene. The mutation site has been shown with white arrows and squares. The prediction of amino-acid transition has been shown on the gene mutation site. H37Rv and K were used as a reference for comparison.

**Table 1 jcm-11-01927-t001:** Drug resistance data of clinical Mtb isolates.

Patients	Isolates	Type	Drug Resistance Profile	Year of Isolation
P1	21873	MDR	INH, RFP, EMB, RBU, PTO	2014
P2	9199	XDR	INH, RFP, EMB, RBU, SM, KM, AMK, OFX, MFX, PAS, PTO, CS	2010
P3	22018	XDR	INH, RFP, EMB, RBU, SM, KM, AMK, OFX, LEV, PTO, CS	2014
	22033	XDR	INH, RFP, EMB, RBU, SM, KM, AMK, OFX, LEV, PTO, CS	2014
P4	22049	XDR	INH, RFP, EMB, RBU, SM, KM, AMK, OFX, LEV, PTO, CS	2014

AMK, amikacin; CS, cycloserine; EMB, ethambutol; INH, isoniazid; KM, kanamycin; LEV, levofloxacin; MDR, Multi-Drug Resistant; MFX, moxifloxacin; OFX, ofloxacin; PAS, para-aminosalicylic acid; PTO, prothionamide; PZA, pyrazinamide; RBU, rifabutin; RFP, rifampicin; SM, streptomycin; XDR, Extensive-Drug Resistant.

## Data Availability

The data that support the findings of this study are available on request from the corresponding author, Park S. The data are not publicly available due to their containing information that could compromise the privacy of research participants.

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
