# Peer review of "Investigation of Clofazimine Resistance and Genetic Mutations in Drug-Resistant Mycobacterium tuberculosis Isolates"

_jcm, 2022, doi:10.3390/jcm11071927_

Round 1
Reviewer 1 Report
This paper presented the results of clofazimine sensitivity testing of mycobacterium tuberculosis isolates from patients with drug resistant tuberculosis in South Korea. The samples were from a sample bank.
Comments.
- The introduction is a little confused in places. It would be good to talk about the place of clofazimine in current regimens.
- Methods: The details of the resazurin microtitre system could be elaborated upon, including its validation for use with clofazimine.
Author Response
- The introduction is a little confused in places. It would be good to talk about the place of clofazimine in current regimens.
: Thank you for your comment. We added places at lines 36-37 and 46-47 in Introduction.
- Methods: The details of the resazurin microtitre system could be elaborated upon, including its validation for use with clofazimine.
: That’s a good point. We added descriptions of lines 87-91 and 98-102 in Methods. The “validation of RMEA” was added at lines 95-96, and moved sentence for the validation for use with clofazimine.
Reviewer 2 Report
This manuscript reports on a search to obtain clofazimine resistance associated mutations in the Korean MDR and XDR isolates of M. tuberculosis. The authors checked 122 isolates for CFZ resistance, selected 5 isolates with MIC exceeding 1 mg/mL, and analyzed the 3 gene loci (Rv0678c, Rv1979 and pepQ) associated with resistance by PCR amplification and the further sequencing of the PCR products. The resistance associated mutations were found in two of three genes.
As the authors admit in their manuscript (lines 274-276 of Discussion), the modest number of isolates under investigation (5) didn’t allow to find any pattern of mutations. Also, one of five isolates didn’t have the mutations in these loci. To make any conclusions, the collection of RFZ resistant isolates should be widen, and the complete genome sequencing of isolates without mutations in these loci should be done.
1. Introduction
Lines 68-70. The authors have no evidence to use the found mutations as potential molecular markers of CFZ resistance
2 Materials and methods
Line 88 Please clarify the difference between 7H9 and 7H9-S broth.
Why such high upper CFZ concentration was used in REMA analysis, 16mg/mL? CFZ MICs typically range from 0,125 to 2,0 mg/mL (lines 53-54)
3. Discussion. The parts concerning sequencing results (lines 22-235, 236-256, 257-269) are very difficult to read. Information is jumbled (in the middle of the paragraph about Rv0678, Rv1979c appeared, lines 226-227), a lot of repetitions. Also, the whole phrases are just duplicated (e.g., lines 222 and 228-229). The English language needs to be corrected.
Author Response
- Introduction
Lines 68-70. The authors have no evidence to use the found mutations as potential molecular markers of CFZ resistance
: Thank you for your comment. We agree with your opinion. We deleted the sentence of “Our results ~ CFZ resistance”, and added at lines 70-71 of “to improve for diagnosis and treatment of drug-resistant TB patients.
- Materials and methods
Line 88 Please clarify the difference between 7H9 and 7H9-S broth.
: The explanation of the 7H9-S broth was added at lines 89-91.
Why such high upper CFZ concentration was used in REMA analysis, 16mg/mL? CFZ MICs typically range from 0,125 to 2,0 mg/mL (lines 53-54)
: It was just for the convenience of results reading in 96-well plates. We added the reason at line 93.
- Discussion. The parts concerning sequencing results (lines 22-235, 236-256, 257-269) are very difficult to read. Information is jumbled (in the middle of the paragraph about Rv0678, Rv1979c appeared, lines 226-227), a lot of repetitions. Also, the whole phrases are just duplicated (e.g., lines 222 and 228-229). The English language needs to be corrected.
: Thank you for your critical and kind comments. The duplicated sentences at lines 223-229, 238-249, and 265-269 were deleted. The line 240-241 were moved and combined at line 227-228 in revised manuscript. The lines 260-262 moved at lines 244-245 in the revised manuscript.
Round 2
Reviewer 2 Report
The manuscript was changed according to my comments, and can be accepted
Author Response
Thank you for your review of our paper. We made corrections by editing English throughout the entire manuscript.